# Oral Administration of Rhamnan Sulfate from *Monostroma nitidum* Suppresses Atherosclerosis in ApoE-Deficient Mice Fed a High-Fat Diet

**DOI:** 10.3390/cells12222666

**Published:** 2023-11-20

**Authors:** Masahiro Terasawa, Liqing Zang, Keiichi Hiramoto, Yasuhito Shimada, Mari Mitsunaka, Ryota Uchida, Kaoru Nishiura, Koichi Matsuda, Norihiro Nishimura, Koji Suzuki

**Affiliations:** 1Konan Chemical Manufacturing Co., Ltd., Kitagomizuka, Kusu-cho, Yokkaichi 510-0103, Japan; terasawa@konanchemical.co.jp (M.T.); uchida@konanchemical.co.jp (R.U.); nisiura@konanchemical.co.jp (K.N.); matsuda@konanchemical.co.jp (K.M.); 2Faculty of Pharmaceutical Sciences, Suzuka University of Medical Science, Minamitamagaki-cho, Suzuka 513-8670, Japan; hiramoto@suzuka-u.ac.jp; 3Graduate School of Regional Innovation Studies, Mie University, Tsu 514-8507, Japan; liqing@doc.medic.mie-u.ac.jp (L.Z.); nishimura.norihiro@mie-u.ac.jp (N.N.); 4Department of Integrative Pharmacology, Mie University Graduate School of Medicine, Tsu 514-8507, Japan; shimada.yasuhito@mie-u.ac.jp (Y.S.); 319110@m.mie-u.ac.jp (M.M.)

**Keywords:** *Monostroma nitidum*, rhamnan sulfate, atherosclerosis, ApoE-deficient mouse

## Abstract

Oral administration of rhamnan sulfate (RS), derived from the seaweed *Monostroma nitidum*, markedly suppresses inflammatory damage in the vascular endothelium and organs of lipopolysaccharide-treated mice. This study aimed to analyze whether orally administered RS inhibits the development of atherosclerosis, a chronic inflammation of the arteries. ApoE-deficient female mice were fed a normal or high-fat diet (HFD) with or without RS for 12 weeks. Immunohistochemical and mRNA analyses of atherosclerosis-related genes were performed. The effect of RS on the migration of RAW264.7 cells was also examined in vitro. RS administration suppressed the increase in blood total cholesterol and triglyceride levels. In the aorta of HFD-fed mice, RS reduced vascular smooth muscle cell proliferation, macrophage accumulation, and elevation of VCAM-1 and inhibited the reduction of Robo4. Increased mRNA levels of *Vcam1*, *Mmp9*, and *Srebp1* in atherosclerotic areas of HFD-fed mice were also suppressed with RS. Moreover, RS directly inhibited the migration of RAW264.7 cells in vitro. Thus, in HFD-fed ApoE-deficient mice, oral administration of RS ameliorated abnormal lipid metabolism and reduced vascular endothelial inflammation and hyperpermeability, macrophage infiltration and accumulation, and smooth muscle cell proliferation in the arteries leading to atherosclerosis. These results suggest that RS is an effective functional food for the prevention of atherosclerosis.

## 1. Introduction

Inflammation of the vascular endothelium can lead to various pathological conditions, such as arterial and venous thrombosis, atherosclerosis, hypertension, and cancerization [1,2]. During the early stages of vascular inflammation, monocytes accumulate in the inflammatory vascular endothelial cells and transform into macrophages. These cells express tissue factor (TF), which triggers extrinsic blood coagulation [3,4]. Further, inflammatory vascular endothelial cells secrete von Willebrand factor (VWF), which induces platelet aggregation [5]. Lipopolysaccharide (LPS), a cell membrane component of Gram-negative bacteria, stimulates the inflammatory response in animal cells by binding to Toll-like receptor 4. LPS significantly stimulates TF and VWF production by vascular endothelial cells [5,6], promoting blood coagulation and platelet aggregation. In addition, LPS induces the secretion of inflammatory cytokines, such as interleukin (IL)-6 and tumor necrosis factor (TNF)-α, by vascular endothelial cells, as well as the expression of various cell adhesion molecules, such as vascular cell adhesion molecule-1 (VCAM-1) and E-selectin [7]. It also promotes leukocyte migration to the sites of inflammation, further promoting inflammation [8,9]. Therefore, reducing inflammation of vascular endothelial cells is important to prevent intravascular thrombus formation and related vascular diseases.

Atherosclerosis is triggered by the accumulation of monocytes, macrophages, activated T lymphocytes, and other cells at the site of inflammation induced by injury to the vascular endothelium (Ross’s injury hypothesis) [10]. Primary fatty streak lesions result from the accumulation of monocyte- and macrophage-derived foam cells, accompanied by vascular smooth muscle cell migration and proliferation in the intima. Oxidized low-density lipoproteins (LDLs) produced under the endothelium advance atherosclerosis through the dysfunction of vascular endothelial cells, foaming of macrophages, and induction of inflammatory reactions. In particular, atherosclerotic plaques, consisting of a thin fibrous capsule, a large lipid core, and a strong inflammatory response, are prone to rupture because of a sudden increase in blood pressure or abnormal blood flow (shear stress) injury. The rupture of this plaque causes occlusion of the vessel lumen with coagulative thrombus formation. Diabetes, visceral obesity, hypertension, and smoking accelerate the progression of atherosclerosis by enhancing oxidative stress and inflammatory reactions in the vessels.

Apolipoprotein E (ApoE) is a glycoprotein synthesized primarily in the liver and is a component of all lipoproteins, including high-density lipoprotein but not LDL. It functions as a transporter in cholesterol and lipoprotein metabolism and as a ligand for receptors that remove chylomicrons and very low-density lipoprotein (VLDL) remnants. ApoE is also synthesized by monocytes and macrophages in vessels and is considered to have local effects on cholesterol homeostasis and inflammatory reactions in atherosclerotic vessels. It may also function in dietary absorption and biliary excretion of cholesterol. ApoE-deficient (ApoE^−/−^) animals are less efficient at removing cholesterol accumulated in blood vessels and are more likely to develop atherosclerosis. Therefore, the ApoE^−/−^ mouse is the most popular murine model used for atherosclerotic studies [11].

*Monostroma nitidum* is a green alga composed of multiple cell aggregates. These aggregates contain intercellular substances rich in dietary fiber, vitamins, and minerals [12]. One of the key components of these intercellular substances is the soluble dietary fiber rhamnan sulfate (RS). It is composed of straight and branched chains of rhamnose units, approximately 25% of which contain sulfate groups [13,14,15,16]. Recent studies have shown that RS exhibits anticoagulant [6,12,17,18], anti-inflammatory [6,12,19], and antiviral properties against herpes, influenza, and SARS-CoV-2 [13,20,21,22]. Furthermore, clinical research has demonstrated that RS can prevent constipation by modulating gut microbiota [23]. These findings suggested that RS plays a major role in the beneficial health effects of *M. nitidum*. Recently, we showed that the oral administration of RS purified from *M. nitidum* significantly suppresses LPS-induced vascular hyperpermeability, as well as inflammatory damage, to the endothelium and organs of mice, both functionally and morphologically [6,19]. Moreover, Patil et al. reported that RS reduces vascular inflammation and atherosclerotic plaque formation histologically in ApoE^−/−^ female mice that are fed a high-fat diet (HFD) [24].

Therefore, in this study, we aimed to investigate the effects of the oral administration of RS on blood lipids and vascular molecules involved in HFD-induced atherosclerosis in ApoE^−/−^ female mice. The results show that oral administration of RS ameliorated abnormal lipid metabolism and reduced atherosclerotic endothelial inflammation and hyperpermeability, macrophage infiltration and accumulation, and smooth muscle cell proliferation in the arteries of HFD-fed ApoE^−/−^ mice. Furthermore, RS was found to directly inhibit the migration of macrophage-like RAW264.7 cells in vitro.

## 2. Materials and Methods

### 2.1. Purification and Molecular Weight Measurement of RS

RS was purified from a hot water extract of *M. nitidum* using a previously described method [6,21]. The molecular weight of RS was determined with gel permeation chromatography (GPC) using a Shodex RI-71 refractive index detector (Showa Denko K. K., Tokyo, Japan) and a Shodex SB-806M HQ column (8.0 × 300 mm). The column was eluted with 0.2 M KNO_3_ at a flow rate of 1.0 mL min^−1^. A standard curve for quantifying the molecular weight of RS was created using pullulan (polysaccharide consisting of glucose) with different molecular weights as standard substances (Appendix A).

### 2.2. Quantification of RS Components

The RS components were quantified as the sum of their constituent sugars, sulfate groups, and sulfate-group-bound cations. One gram of purified RS was dissolved in 30 mL of water, sonicated for 30 min, and centrifuged (1600× *g*, 10 min). The obtained supernatant was further centrifuged using a centrifugal ultrafiltration device (molecular weight cut off: 3000; Nihon Pall Ltd., Tokyo, Japan), and fractions with a molecular weight of 3000 or more were collected and used for quantification.

#### 2.2.1. Standard Materials

Rhamnose monohydrate (Tokyo Chemical Industry Co., Ltd., Tokyo, Japan), glucuronic acid (Thermo Fisher Scientific, Waltham, MA, USA.), glucose (Kanto Chemical Co., Inc., Tokyo, Japan), anhydrous galactose (Tokyo Chemical Industry Co.), and xylose (Tokyo Chemical Industry Co.), weighing 100 mg each, were dissolved in pure water and serially diluted to obtain standard solutions. Anion-mixed standard solution IV (Kanto Chemical Co., Inc.) and ICP standard solution D (Kanto Chemical Co., Inc.) were also used as standard solutions.

#### 2.2.2. Measurement of Constituent Sugars

The samples were hydrolyzed using 0.5 mol/L sulfuric acid in an autoclave at 120 °C for 60 min. After adding 1 mL of internal standard solution (galactosamine hydrochloride 10 mg/mL), 20 µL of the mixture was transferred to another tube. Then, 80 µL of 0.5 mol/L sodium hydroxide solution and 100 µL of PMP solution (3-methyl-1-phenyl-5-pyrazolone 88 mg/mL in methanol) were added, and the mixture was incubated at 60 °C for 30 min. After adding 1 mL of water, 50 µL of 0.5 mol/L sulfuric acid, and 2 mL of toluene, the supernatant was removed after centrifugation at 1500× *g* for 10 min. Then, the lower layer was stirred, 2 mL toluene was added, and the mixture was centrifuged again. This washing procedure was performed five times by discarding the supernatant layer after each washing. Finally, the lower layer was filtered through a 0.45 μm pore size membrane filter and used as the test solution. The test and standard measurement solutions were analyzed with high-performance liquid chromatography (instrument: high-performance liquid chromatography Chromaster (Hitachi High-Tech Corporation, Tokyo, Japan); detector: UV detector; column: TSKgel ODS-80Ts 5 μm, 2.0 mm I.D. × 250 mm (Tosoh Co., Tokyo, Japan); column temperature: 35 °C; eluent A: water, eluent B: phosphate buffer solution, eluent C: mixed solvents; flow rate: 0.15 mL/min; injection volume: 5 μL; detection wavelength: 245 nm; mobile phase measured under gradient conditions were as follows: 0 min: 45% (A), 33% (B), and 22% (C); 35 min: 30% (A), 42% (B), and 28% (C); 37 min: 25% (A), 45% (B), and 30% (C); 39 min: 45% (A), 33% (B), and 22% (C); and 48 min: 45% (A), 33% (B), and 22% (C)).

#### 2.2.3. Measurement of Sulfate Groups

The sample was hydrolyzed using 2 mol/L hydrochloric acid in an autoclave at 100 °C for 4 h, neutralized with sodium hydroxide, and diluted 1000 times with water. The diluted solution was filtered through a membrane filter with a pore size of 0.45 μm, and the filtrate was used as the test solution. The test and standard solutions were analyzed using ion chromatography (instrument: Dionex ICS-2100 (Thermo Fisher Scientific, Waltham, MA, USA); detector: conductivity detector; column: Dionex IonPacAS19 (Thermo Fisher Scientific); column temperature: 35 °C; eluent: Dionex EGCIII KOH (Thermo Fisher Scientific); flow rate: 1.0 mL/min; injection volume: 250 μL; mobile phase gradient conditions were as follows: 0 min, 5 mM; 0.5 min, 5 mM; 8 min, 20 mM; 11 min, 40 mM; 13.5 min, 50 mM; 15 min, 50 mM; and 20 min, 5 mM).

#### 2.2.4. Measurement of Sulfate-Bound Cations

The content of the sulfate-bound cations in the sample comprised the total content of sodium, potassium, magnesium, and calcium. For quantitation of the sodium and potassium, 100 mL of 1% hydrochloric acid was added to 10 mL of the sample solution, shaken for 30 min, filtered, and further diluted 10-fold with 1% hydrochloric acid to prepare a test solution. For quantitation of the magnesium and calcium, the sample solution was placed in a crucible, prewashed on a hot plate, heated to 550 °C in an electric furnace, and incinerated. Then, 3 mL of hydrochloric acid was added to the ash (1 + 1) and evaporated on a hot plate. Thereafter, 20 mL of 1% hydrochloric acid was added to it, covered with a watch glass, and heated at 100 °C for 1 h. Subsequently, the solution was filtered, diluted to 20 mL with 1% hydrochloric acid, further diluted 100 times with 1% hydrochloric acid, and then used as the test solution. The test and standard solutions were analyzed using inductively coupled plasma emission spectrometry (ICP-OES; instrument: Optima 8300 ICP-OES (PerkinElmer, Waltham, MA, USA); RF power: 1300 W; measurement wavelengths and direction of plasma observations were as follows: sodium: 588.995, axial direction; potassium: 766.490, axial direction; magnesium: 280.271, axial direction; and calcium: 393.366, lateral direction.

### 2.3. Animal Experiments

ApoE^−/−^ female mice (B6.KOR/StmSlc-Apoeshl strain), spontaneously hyperlipidemic Japanese wild mice (KOR) with ApoE deficiency were purchased from SLC Japan Inc. (Hamamatsu, Japan), and housed under a 12 h light/dark cycle at the Institute of Laboratory Animals at Mie University, Tsu, Japan. Twelve-week-old ApoE^−/−^ mice were randomly assigned to four groups of six mice each. Mice in the (1) normal diet (ND) group were fed a normal diet (CLEA Rodent Diet CE-7; CLEA Japan, Tokyo, Japan); mice in the (2) ND + RS group were fed an ND containing 0.1% RS (*w/w*) (equivalent to 150 mg/kg body weight/day); mice in the (3) HFD group were fed a high-fat rodent diet containing 1.25% cholesterol (40% energy from fat; D12108C; Research Diets, New Brunswick, NJ, USA); and mice in the (4) HFD + RS group were fed a HFD containing 0.1% RS (*w/w*) (equivalent to 150 mg/kg body weight/day). The duration of the experiment was 12 weeks. Body weight, fasting blood glucose levels, and food intake were measured once weekly. The experimental schedule is shown in Appendix A.

The mice were euthanized with anesthesia with isoflurane (Pfizer, Pearl River, NY, USA) and subjected to laparotomy. Blood was collected, and plasma levels of triglyceride (TG) and total cholesterol (TCHO) were measured using Wako L-type TG and Wako L-type TCHO assay kits (Fujifilm Wako Pure Chemicals, Osaka, Japan), respectively, according to the manufacturer’s protocol. Liver and abdominal aorta tissues were dissected for subsequent histological and quantitative real-time PCR (qPCR) analyses.

### 2.4. Quantitative Real-Time PCR

Total RNA was extracted from liver and abdominal aorta tissues and purified using the QIAGEN RNeasy Miniprep Kit (QIAGEN, Hilden, Germany), according to the manufacturer’s instructions. cDNA was synthesized from 500 ng of total RNA using the ReverTra Ace qPCR RT Kit (Toyobo, Osaka, Japan), according to the manufacturer’s instructions. qPCR was performed using the Power SYBR Green Master Mix (Applied Biosystems, Foster City, CA, USA) and the ABI StepOnePlus Real-Time PCR System (Applied Biosystems), according to the manufacturer’s instructions. The relative mRNA levels were determined using hypoxanthine guanine phosphoribosyltransferase 1 (*Hprt1*) [25] as the endogenous control gene. The primer sequences used for the PCR amplification are listed in Appendix A. The primer sequences used to determine the mRNA expression levels were prepared from the following genes: matrix metallopeptidase 2 (*Mmp2*) [26], *Mmp9* [27], sterol regulatory element-binding transcription factor 1 (*Srebf1*) [28], intercellular adhesion molecule 1 (*Icam1*) [27], and vascular cell adhesion molecule 1 (*Vcam1*) [29].

### 2.5. Immunohistochemistry

Liver and aorta specimens were fixed in 4% phosphate-buffered paraformaldehyde, embedded in frozen Tissue Tek O.C.T. C Compound (Sakura Finetek, Tokyo, Japan) and sliced into 5 μm thick sections. The sections were stained with antibodies for immunohistochemical analysis, as previously described [30]. The liver and aorta specimens were incubated with a primary antibody, such as rat monoclonal anti-F4/80 (a marker of macrophages) (1:100; Bio-Rad Laboratories, Hercules, CA, USA), rabbit monoclonal anti-alpha smooth muscle actin (αSMA) (1:100; Arigo Biolaboratories, Hsinchu City, Taiwan, ROC), rabbit monoclonal anti-platelet-derived growth factor receptor β (PDGFRβ) (1:100; Cell Signaling Technology, Danvers, MA, USA), rabbit monoclonal anti-VCAM-1 (1:100; Abcam, Cambridge, MA, USA), or mouse monoclonal anti-Roundabout 4 (Robo4) (an endothelium-specific vascular permeability inhibitory molecule) (1:100; Santa Cruz Biotechnology, Santa Cruz, CA, USA). The sections were subsequently incubated with fluorescein- or tetramethylrhodamine isothiocyanate-conjugated anti-rabbit, anti-rat, or anti-mouse secondary antibodies (1:30; Dako Cytomation, Glostrup, Denmark). The expressions of F4/80, αSMA, PDGFRβ, VCAM-1, and Robo4 were evaluated immunohistochemically using fluorescence microscopy. The fluorescence intensities were quantified using fluorescence microscopy wth ImageJ software ver. 1.53 (National Institutes of Health, Bethesda, MD, USA).

### 2.6. Migration Assay

A transwell migration assay (Corning Inc., Corning, NY, USA) was performed using mouse macrophage-like RAW264.7 cells [31] obtained from RIKEN (RCB0535, Tsukuba, Japan). Dulbecco’s Modified Eagle′s Medium low glucose (Sigma-Aldrich, St. Louis, MO, USA), supplemented with 10% fetal bovine serum (Sigma Chemical Company, St. Louis, MO, USA) and 1% penicillin-streptomycin (Fujifilm Wako Pure Chemicals), was used for the cell culture. The cells (1 × 10^5^) were suspended in a serum-free medium and seeded in the upper chamber. The lower chamber was filled with 500 μL of whole serum culture medium. RS (10 or 100 μg/mL) was added to the upper or lower chambers and incubated for 4 or 24 h. The medium was then removed from the upper and lower chambers, and the chambers were washed twice with phosphate-buffered saline. The cells migrated from the upper chamber to the lower chamber through a membrane filter. The untreated cells were used as controls. The nonmigrating cells in the upper chamber were scraped off the top surface of the membrane filter using a cotton swab. The cells that migrated to the underside of the membrane filter were fixed with 4% formaldehyde solution, stained with Hoechst/4’,6-diamidino-2-phenylindole (DAPI), and quantified using light microscopy.

### 2.7. Statistical Analysis

All results are shown as the mean ± standard deviation. Multiple comparisons between groups were analyzed using two-way analysis of variance (ANOVA) followed by Tukey’s post hoc test using GraphPad Prism software version 10 (GraphPad Software, San Diego, CA, USA). The migration assay was analyzed using one-way ANOVA with a Dunnet’s post-test using Prism software version 10 (GraphPad Software). Statistical significance was set at *p* ≤ 0.05. Pearson correlation values (*r*) between plasma TCHO or TG levels and aortic protein expression levels analyzed using immunohistochemistry were calculated using GraphPad Prism version 10 (GraphPad Software).

## 3. Results

### 3.1. Chemical Characteristics of RS Isolated from M. nitidum

After extracting RS from *M. nitidum*, we analyzed its chemical properties. The analyses revealed a single major peak on the GPC column, with weight-average (*M_w_*), number-average (*M_n_*), and z-average (*M_z_*) molecular weights of 148, 28, and 844 kDa, respectively (Appendix A). The RS content was defined as the total amount of constituent monosaccharides, sulfate groups, and cations bound to the sulfate group. The quantity of the constituent monosaccharides, sulfate groups, and cations was 55, 32, and 6.9 g per 100 g, respectively. This yielded an RS content of 94 g per 100 g (purity 94%). The monosaccharide composition of RS included glucuronic acid, rhamnose, glucose, galactose, and xylose, with a content of 3.2, 49, 0.7, <0.4, and 2.3 g per 100 g, respectively. The cation composition of RS comprised sodium, potassium, magnesium, and calcium, with a content of 0.12, <0.05, 0.54, and 6.2 g per 100 g, respectively. These data indicate that RS is a rhamnose-dominant sulfated polysaccharide, consistent with previous studies [15,32].

### 3.2. RS Improved the Lipid Profile of HFD-Fed ApoE^−/−^ Mice

The effect of orally administered RS on the body weight and lipid metabolism of ApoE^−/−^ mice was investigated. A significant increase in body weight (*p* < 0.001) was observed in mice in the HFD-fed group compared to mice in the ND-fed group. Furthermore, when comparing mice in the HFD + RS-fed and HFD-fed groups, a significant increase in body weight (*p* < 0.05) was observed in the RS-fed mice (Appendix A). These data indicate that RS did not adversely affect the growth of mice in the HFD-fed group. Furthermore, RS did not affect food intake during feeding experiments (Appendix A).

Figure 1A shows the plasma TCHO levels in both the ND- and HFD-fed ApoE^−/−^ mice treated with or without RS. The plasma TCHO levels were significantly lower (*p* < 0.05) in mice in the RS-treated ND-fed group compared to those in mice in the RS-untreated ND-fed group and more significantly lower (*p* < 0.01) in mice in the HFD + RS-fed group compared to those in mice in the HFD-fed group. This result suggests that RS suppressed the increase in cholesterol production in the ND- and HFD-fed groups of ApoE^−/−^ mice. To confirm the suppressive effect of RS on TCHO production in mice, the effect of RS on *Srebp1* mRNA expression in the aorta of ND- or HFD-fed ApoE^−/−^ mice was analyzed using qPCR. As shown in Figure 1B, *Srebp1* mRNA levels were significantly higher in the aorta of HFD-fed ApoE^−/−^ mice than those in the ND-fed mice and significantly lower (*p* < 0.05) in the aorta of HFD + RS-fed mice than those in the HFD-fed mice. These data suggest that RS suppressed *Srebp1* mRNA expression in the aorta, which may lead to decreased plasma cholesterol levels.

Figure 1C shows the effect of RS administration on the plasma TG levels in the HFD-fed ApoE^−/−^ mice. The plasma TG levels were significantly lower (*p* < 0.001) in the HFD + RS-fed group than those in the HFD-fed group of the ApoE^−/−^ mice, indicating that RS strongly suppressed increased TG production in the HFD-fed group of ApoE^−/−^ mice. These data suggest that RS improved the lipid profile of the HFD-fed ApoE^−/−^ mice.

### 3.3. RS Reduced VCAM-1 Expression in the Liver and Aorta of HFD-Fed ApoE^−/−^ Mice

As shown in Figure 2A–C, the expression of VCAM-1 in the liver and aorta was significantly higher (*p* < 0.001) in the HFD-fed group than that in the ND-fed group. The VCAM-1 expression in both the liver and aorta in the HFD + RS-fed group was significantly lower (*p* < 0.001) than that in the HFD-fed group. Figure 2D shows the *Vcam1* mRNA expression in the aorta of both the ND- and HFD-fed ApoE^−/−^ mice. The *Vcam1* mRNA levels were significantly higher (*p* < 0.05) in the HFD-fed group than that in the ND-fed group and were significantly lower (*p* < 0.01) in the HFD + RS-fed group than that in the HFD-fed group. This indicates that RS significantly suppressed the *Vcam1* mRNA expression in the aorta of the HFD-fed ApoE^−/−^ mice.

Figure 3A–C show the expression of ICAM-1 in the liver and aorta of both the ND- and HFD-fed ApoE^−/−^ mice. The expression of ICAM-1 in both tissues was low for both ND- and HFD-fed ApoE^−/−^ mice. The ICAM-1 expression was not affected by RS administration in either tissue type. Figure 3D shows the *Icam1* mRNA expression in the aorta of the ND- and HFD-fed ApoE^−/−^ mice, indicating that RS did not affect the *Icam1* mRNA expression in the aorta of the ND- or HFD-fed ApoE^−/−^ mice.

### 3.4. RS Reduced Macrophage Accumulation in the Liver and Aorta of HFD-Fed ApoE^−/−^ Mice

Macrophages that accumulate at sites of inflammation differentiate into tissue-specific cells, such as aortic foam cells after oxide LDL uptake and Kupffer cells in the liver [33]. Next, we evaluated whether RS administration suppressed infiltration of macrophages into the liver and aorta, using an antibody against the 160 kDa cell surface glycoprotein F4/80, a macrophage-specific marker.

Figure 4A–C show that the expression of F4/80 in both the liver and aorta was significantly higher (*p* < 0.001) in the HFD-fed group than that in the ND-fed group. Further, the F4/80 expression in the liver and aorta was significantly lower (*p* < 0.001) in the HFD + RS-fed group than that in the HFD-fed group. These data suggest that RS administration suppressed macrophage accumulation and proliferation in both the liver and aorta of the ApoE^−/−^ mice, regardless of ND or HFD feeding.

### 3.5. RS Reduced PDGFRβ Expression in the Aorta of HFD-Fed ApoE^−/−^ Mice

The effect of RS treatment on the expression of PDGFRβ in the liver and aorta of the ND- or HFD-fed ApoE^−/−^ mice was determined using immunohistochemistry. As shown in Figure 5A–C, the expression level of PDGFRβ was lower in the liver of mice in the ND group, but the PDGFRβ expression levels in the aorta tended to decrease in the mice in the ND + RS-fed group compared to that in the mice in the ND-fed group. The expression of PDGFRβ in both the liver and aorta was significantly higher in the HFD-fed group than that in the ND-fed group. Although the PDGFRβ expression did not decrease in the liver, it was significantly lower (*p* < 0.001) in the aorta of the mice in the HFD + RS-fed group than that in the HFD-fed group. This result indicates that RS administration suppresses PDGFRβ-mediated atherosclerosis promotion.

### 3.6. RS Reduced αSMA Expression in the Liver and Aorta of HFD-Fed ApoE^−/−^ Mice

Vascular smooth muscle cells express αSMA, contributing to vascular motility and contraction. The effect of RS on the αSMA expression in the liver and aorta of both the ND- and HFD-fed ApoE^−/−^ mice was studied. As shown in Figure 6A–C, the expression level of αSMA-positive cells was lower in the liver and aorta of ND-fed ApoE^−/−^ mice regardless of RS administration. The expression level of αSMA was significantly higher (*p* < 0.001) in both the liver and aorta of mice in the HFD-fed group than that in the ND-fed group. The αSMA expression level was significantly lower (*p* < 0.001) in the HFD + RS-fed group than that in the HFD-fed group. These results indicate that RS administration suppresses vascular smooth muscle cell proliferation in the liver and aorta, suggesting that RS ameliorates atherosclerosis development.

### 3.7. Correlation between Plasma TCHO or TG Levels and Aortic Atherosclerotic Molecule Levels in ND- or HFD-Fed ApoE^−/−^ Mice with or without RS Administration

From the aforementioned results, plasma TCHO and TG levels, which were elevated in the HFD-fed ApoE^−/−^ female mice, significantly decreased in the RS-fed group, and the expression of aortic atherosclerotic molecules, VCAM-1, F4/80, PDGFRβ, and αSMA, except for ICAM-1, significantly decreased in the RS-fed group. Therefore, we next analyzed the correlation between the effects of the oral administration of RS on changes in plasma TCHO and TG levels and changes in the level of each aorta-expressed molecule. Table 1 shows the Pearson correlation coefficient (*r*) of each aorta-expressed molecule in relation to the plasma TCHO and TG levels. The data show that PDGFRβ (*p* < 0.05), αSMA (*p* < 0.05), VCAM-1, and ICAM-1 have higher correlation values with plasma TCHO than those with plasma TG, and that the correlation of the macrophage-specific molecule F4/80 (*p* < 0.05) is significantly higher with plasma TG than that with plasma TCHO. Furthermore, the effect of orally administered RS on the relationship between plasma TCHO or TG levels and the expression level of each aortic molecule in the ND- or HFD-fed ApoE^−/−^ mice is shown in Appendix A. The results show that the oral administration of RS to HFD-fed ApoE^−/−^ mice caused changes in the expression levels of VCAM-1, F4/80, and PDGFRβ and that these changes are closely correlated with changes in the plasma TCHO levels. Meanwhile, changes in the F4/80 expression levels were closely correlated with changes in the plasma TG levels. In addition, the expression levels of ICAM-1 were not associated with the plasma TCHO or TG levels.

### 3.8. Effect of RS on the Expression of Robo4 in the Liver and Aorta of HFD ApoE^−/−^ Mice

As shown in Figure 7A–C, no significant change was observed in the expression of Robo4 in the liver and aorta of the ND-fed mice, regardless of whether RS was orally administered. In contrast, the expression of Robo4 was reduced by more than half in the liver and was significantly lower (*p* < 0.01) in the aorta of the HFD-fed group than in the ND-fed group. Moreover, Robo4 expression was significantly higher (*p* < 0.01) in both the liver and aorta of the RS + HFD-fed group than in the HFD-fed group. These data suggest that Robo4 expression is highly reduced in the blood vessels of HFD-fed ApoE^−/−^ mice, resulting in higher infiltration of monocytes and macrophages into the arterial media. Furthermore, administration of RS to HFD-fed ApoE^−/−^ mice suppressed the downregulation of Robo4, resulting in a significant decrease in the infiltration of monocytes/macrophages into the arterial media.

### 3.9. RS Reduced Mmp9 mRNA Expression in the Aorta of ApoE^−/−^ Mice

The effect of RS administration on *Mmp2* and *Mmp9* mRNA expression in the aorta of the ND- or HFD-fed ApoE^−/−^ mice was analyzed using qPCR. Figure 8A shows that the *Mmp2* mRNA expression in the aorta of the HFD-fed mice was slightly higher than that in the aorta of the ND-fed mice, but this increase was not affected by RS administration. In contrast, as shown in Figure 8B, the *Mmp9* mRNA expression tended to be higher in the aorta of HFD-fed mice than that in the aorta of the ND-fed mice, and this increase was significantly suppressed (*p* < 0.05) with RS administration. These results suggest that RS prevents atherosclerosis by suppressing MMP-9 production in ApoE^−/−^ mice.

### 3.10. RS Directly Inhibited the Migration of Mouse Macrophage-Like Cells In Vitro

The direct effect of RS on macrophage migration was investigated using an in vitro transmigration assay with RAW264.7 cells. Figure 9 shows that when the incubation time of the RS-untreated cells increased from 4 to 24 h, the number of migrated cells, that is, the number of stained cells on the underside of the membrane filter, increased. The migration of cells treated with RS (more than 10 µg/mL) was significantly inhibited (*p* < 0.001) after 4 h regardless of whether RS was added to the upper or lower chamber. These results suggested that RS may act directly on macrophages to reduce their migratory ability, leading to the suppression of atherosclerosis.

## 4. Discussion

Atherosclerosis is thought to be caused by macrophages invading the vascular tunica media at the site of inflammation, triggered by various factors in the blood vessels. These macrophages take up LDL cholesterol from the blood and enlarge, resulting in increased inflammation. This causes further migration of macrophages into proliferating smooth muscles to form plaques, resulting in enlargement of the vascular tunica media. Large amounts of cholesterol or neutral fat in LDL and VLDL cause atherosclerosis [34].

During the early stages of atherosclerosis, endothelial dysfunction and lipid metabolism induce endothelial cell activation and permeability. Oxidized LDL then accumulates on endothelial cells, and monocytes differentiate into macrophages. LDL accumulation leads to increased vasoconstriction, smooth muscle proliferation, platelet aggregation, leukocyte adhesion, and the release of inflammatory cytokines [34]. TCHO and TG are known to play key roles in atherosclerosis progression.

Our previous mouse experiments and clinical studies showed that oral administration of *M.-nitidum*-derived RS lowered blood cholesterol levels [12,35] and that RS attenuated LPS-induced vascular inflammation and organ damage in mice [19]. In this study, we showed that orally administered RS significantly reduced the increased TCHO and TG levels in the blood of HFD-fed ApoE^−/−^ mice and suppressed *Srebp1* mRNA expression in arterial vessels, indicating that RS suppresses cholesterol production in the liver and blood vessels, thereby lowering blood LDL levels and reducing atherosclerosis. SREBP-1 plays an important role in the production and accumulation of lipids and cholesterol in the liver. Its mRNA is progressively upregulated in fat streaks and fibrous lipid plaques [36].

Atherosclerosis is believed to proceed through the binding of monocytes and macrophages in inflammatory lesions, as well as the binding of leukocytes. Thus, we investigated the effect of RS on the expression of VCAM-1, which selectively binds to monocytes and macrophages and stimulates macrophage proliferation and accumulation [37] and ICAM-1, which selectively binds to leukocytes and promotes inflammation, in the liver and aorta of ApoE^−/−^ mice fed an ND or HFD. RS administration decreased both the levels of VCAM-1 and its mRNA. VCAM-1 is necessary for macrophage migration and proliferation and causes macrophage accumulation in the plaques and increased plaque instability [38]. In the current study, there was no increase in the ICAM-1 expression levels in the HFD-fed ApoE^−/−^ mice, probably because the experiment was conducted for 12 weeks, which could be too early for the expression of ICAM-1. It has been reported that increased expression of both adhesion molecules VCAM-1 and ICAM-1 is found in ApoE^−/−^ mice [39]. However, experiments using VCAM-1 and ICAM-1 knockout mice have reported that VCAM-1 plays a major function compared to ICAM-1 in the early stage of atherosclerosis [37]. RS administration suppresses the expression of VCAM-1 and its mRNA but not ICAM-1 and its mRNA. This suggests that RS administration has anti-inflammatory effects in the early stages of atherosclerosis and that RS significantly suppresses macrophage accumulation and proliferation in the inflammatory lesions of the aorta of HFD-fed ApoE^−/−^ mice. Indeed, data on the effect of RS on F4/80 expression in the HFD-fed ApoE^−/−^ mice show that RS administration significantly suppressed macrophage accumulation and proliferation in both the liver and aorta of the mice.

PDGFRβ, a member of the PDGFR family of membrane receptors with tyrosine kinase activity, is overexpressed in atherosclerotic lesions, and its signaling enhances local inflammation and synergizes with hypercholesterolemia to promote atherosclerotic arteries [40]. Indeed, this study showed that an increased expression of PDGFRβ was observed in the liver and aorta of HFD-fed ApoE^−/−^ mice, and the increased expression of PDGFRβ in the aorta was significantly reduced by RS administration. Similarly, the increased expression of αSMA in the liver and aorta of the HFD-fed ApoE^−/−^ mice was significantly suppressed by RS administration. These data suggest that RS ameliorates atherosclerosis progression.

Pearson correlation coefficient data for each aorta-expressed molecule against changes in the plasma TCHO and TG levels caused by oral administration of RS show that changes in the vascular cell molecules (PDGFRβ, αSMA, VCAM-1, and ICAM-1) correlated more with changes in the plasma TCHO than those with plasma TG. However, changes in the macrophage-specific molecule F4/80 showed a higher correlation with changes in plasma TG than those with plasma TCHO. Saja et al. demonstrated that hypertriglyceridemia promotes monocytes extravasation and tissue macrophage accumulation [41]. Since high levels of TG-rich lipoproteins can promote macrophage formation into foam cells and plaque formation, lowering TG levels by RS administration may lead to inhibition of plaque formation.

It has been reported that the expression of both MMP-9 and MMP-2 is increased in HFD-fed ApoE^−/−^ mice with the development of atherosclerosis [42]. MMPs are highly expressed in atherosclerotic plaques, and proteolytic disruption of plaques by MMPs is involved in the progression of atherosclerosis and the development of acute coronary syndromes [42]. In this study, RS administration suppressed the increase in MMP-9 expression. It should be noted that MMP-9 expression was observed at 15 weeks of age, whereas MMP-2 expression was detected after 20 weeks. Therefore, the results of this study suggest that MMP-9, but not MMP-2, is expressed during the early stages of atherogenesis.

Nuclear factor-κB (NF-κB) is a key molecule involved in the regulation of inflammatory genes induced by abnormal shear stress and several pro-atherosclerotic factors [43]. The p50/p65 heterodimer that composes NF-κB binds to the promoters of atherogenic genes of ICAM-1, VCAM-1, MCP-1, MMPs, and cathepsins [44,45,46], which promote macrophage recruitment and smooth muscle cell growth and migration. NF-κB also stimulates the expression of coagulation factor TF and platelet aggregation factor VWF, which lead to thrombus formation [47]. Recently, it was reported that RS directly binds to fibro-blast growth factor 2, PDGF-BB, and NF-κB and shows inhibitory effects on inflammation via regulation of the NF-κB pathway [24], probably via regulation of the expression of VCAM-1, MMP-9, and others. RS is a robust inhibitor of NF-κB pathway activation, which is recognized for its involvement in proinflammatory signaling in atherosclerosis.

Our previous studies showed that RS strongly suppressed the inflammation of cultured human umbilical vein endothelial cells by increasing the expression of TF and VWF induced by thrombin, TNF-α, and LPS [6]. Further, orally administered RS significantly inhibited vascular endothelial damage, including the disappearance of glycosaminoglycans from the endothelial surface and inflammation in many organs in LPS-treated mice [19]. In addition, the vascular hyperpermeability state induced by intraperitoneal administration of LPS was suppressed in mice that were orally fed RS [19]. In contrast, it has been reported that the increased permeability caused by LPS administration is suppressed in mice overexpressing the vascular permeability inhibitory molecule Robo4, which is produced specifically in vascular endothelial cells and suppresses inflammation-induced hyperpermeability by stabilizing vascular endothelial cadherin at cell junctions [48]. This study showed that Robo4 expression was significantly decreased in the aorta of ApoE^−/−^ mice with atherosclerosis induced by HFD feeding and that oral administration of RS significantly suppressed the decrease in Robo4 expression.

In this study, we found that RS directly inhibited the migration of mouse macrophage-like RAW264.7 cells in vitro. This finding suggests that RS absorbed into the body directly acts on monocytes and macrophages and may suppress their infiltration into vascular media that causes atherosclerosis.

Finally, on the basis of the current results, it can be estimated that consuming about 2 g of dried *M. nitidum* per day by a person weighing 60 kg may exert an anti-atherosclerotic effect, as calculated from the human equivalent dose based on the body surface area (12.3 for mouse) [49] and the RS content of *M. nitidum* being approximately 40% (internal data, unreported). From previous studies on fluorescein-isothiocyanate-labeled RS, it is known that RS passes through M cells and enters the Peyer’s patches in the gastrointestinal tract after 30 min of oral administration [21]. However, it is unclear what substances RS or its metabolites bind to when consumed and whether they are involved in the expression of genes and proteins that are, in turn, involved in the expression of RS-dependent/RS-specific activities. This remains to be investigated in the future. Such analyses require various techniques to analyze RS and its effects of the body, for example, molecular docking prediction, analysis using RS-specific antibodies or radioisotope-labeled RS, and experiments, such as pull-down assays and affinity chromatography, should be conducted, as performed by Laeliocattleya et al. [50] in their research on fucoidan.

Therefore, the data obtained in this study suggest that RS attenuates atherosclerosis by improving lipid metabolism, inhibiting the expression of inflammatory substances, reducing vascular hyperpermeability, and suppressing the migration of macrophages and proliferation of vascular smooth muscle cells. RS is considered a functional food that is effective in preventing atherosclerosis.

## 5. Conclusions

Oral administration of RS ameliorated the abnormal lipid metabolism in HFD-fed ApoE^−/−^ mice and reduced atherosclerotic and endothelial inflammation, vascular hyperpermeability, macrophage infiltration and accumulation, and smooth muscle cell proliferation in the arteries. Our findings suggest that RS is an effective functional food for the prevention of atherosclerosis.

## Figures and Tables

**Figure 1 cells-12-02666-f001:**
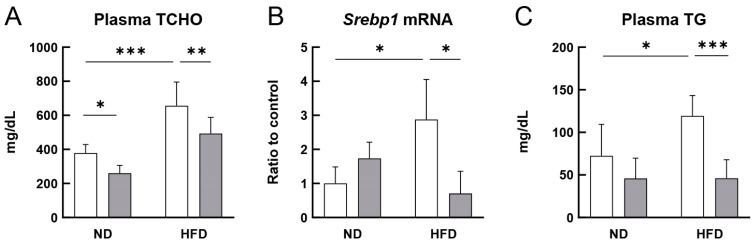
Effects of rhamnan sulfate (RS) on the plasma lipid parameters and sterol regulatory element-binding protein 1 (*Srebp1*) mRNA expression. Apolipoprotein E-deficient (ApoE^−/−^) mice were fed either a normal diet (ND) or a high-fat diet (HFD) for 12 weeks. The groups were divided into two groups: one group not treated with RS (open columns) and the other with 0.1% RS (gray columns). Graphical representation of (**A**) plasma levels of total cholesterol (TCHO), (**B**) *Srebp1* mRNA levels in the aorta quantified using quantitative real-time PCR (qPCR), and (**C**) plasma levels of triglyceride (TG) in each group of mice. Data are presented as the mean ± standard deviation (SD). * *p* < 0.05, ** *p* < 0.01, and *** *p* < 0.001 indicate significant differences between each group using two-way ANOVA (n = 5 or 6 for lipid parameter assays; n = 3 for qPCR).

**Figure 2 cells-12-02666-f002:**
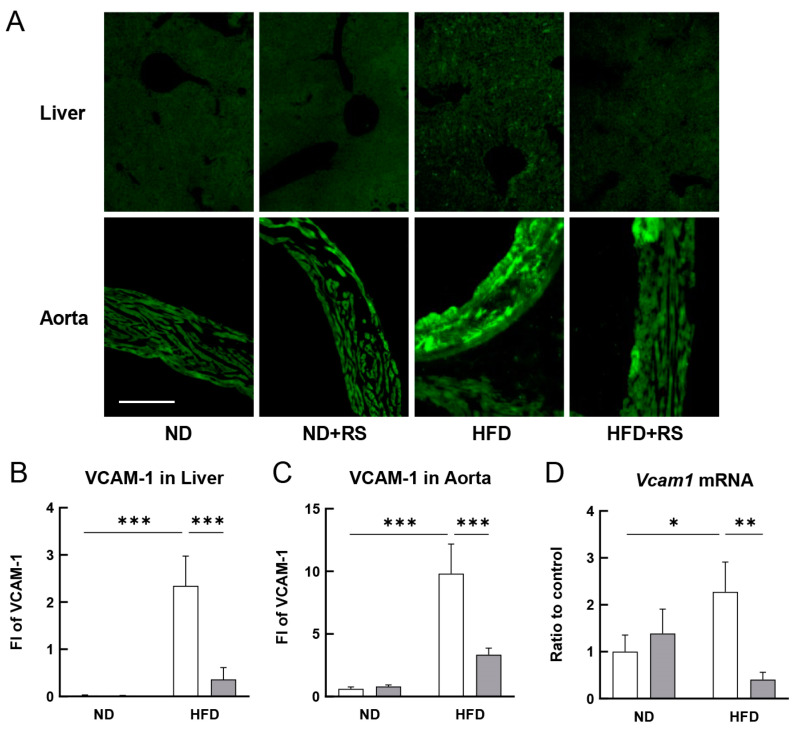
Effect of RS on vascular cell adhesion molecule-1 (VCAM-1) expression in the liver and aorta of ApoE^−/−^ mice fed an ND or a HFD supplemented with or without RS. (**A**) Immunohistochemical analysis of VCAM-1 in the liver and aorta of mice in the ND, ND + RS, HFD, and HFD + RS groups. Scale bar = 100 μm. Graphical representation of the fluorescence intensity (FI) of VCAM-1 in the (**B**) liver and (**C**) aorta of mice in the ND and HFD groups. (**D**) Graphical representation of the *Vcam1* mRNA levels in the aorta of mice in the ND and HFD groups quantified using qPCR. Open and gray columns indicate the RS-untreated and RS-treated groups, respectively. Data are shown as the mean ± SD. * *p* < 0.05, ** *p* < 0.01, and *** *p* < 0.001 indicate significant differences between each group using two-way ANOVA (n = 4 for immunohistochemistry; n = 3 for qPCR).

**Figure 3 cells-12-02666-f003:**
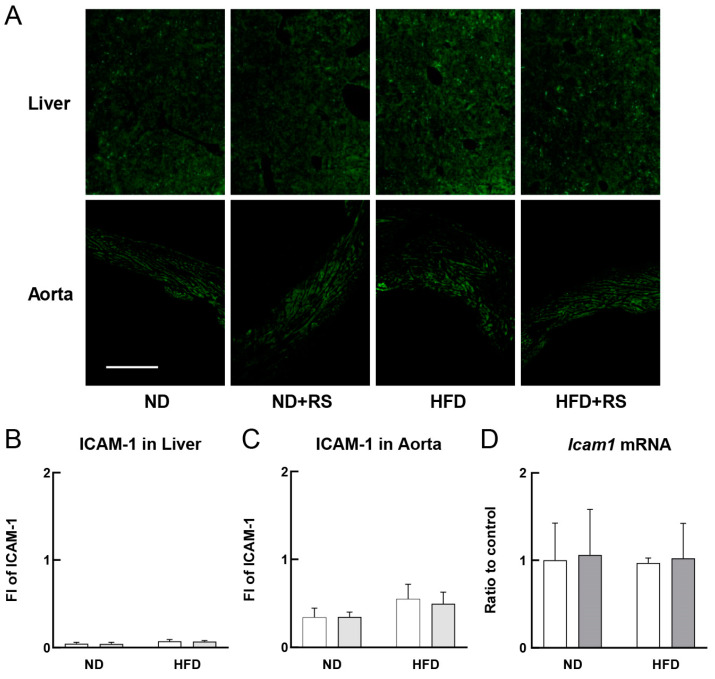
Effects of RS on intercellular adhesion molecule-1 (ICAM-1) expression in the liver and aorta of ApoE^−/−^ mice fed an ND or HFD treated with or without RS. (**A**) Immunohistochemical analysis of ICAM-1 in the liver and aorta of mice in the ND, ND + RS, HFD, and HFD + RS groups. Scale bar: 100 μm. The columns show the FI of ICAM-1 in the (**B**) liver and (**C**) aorta. (**D**) *Icam1* mRNA level in the aorta quantified using qPCR. Open and gray columns indicate the RS-untreated and RS-treated groups, respectively. Data are shown as the mean ± SD. (n = 4 for immunohistochemistry; n = 3 for qPCR).

**Figure 4 cells-12-02666-f004:**
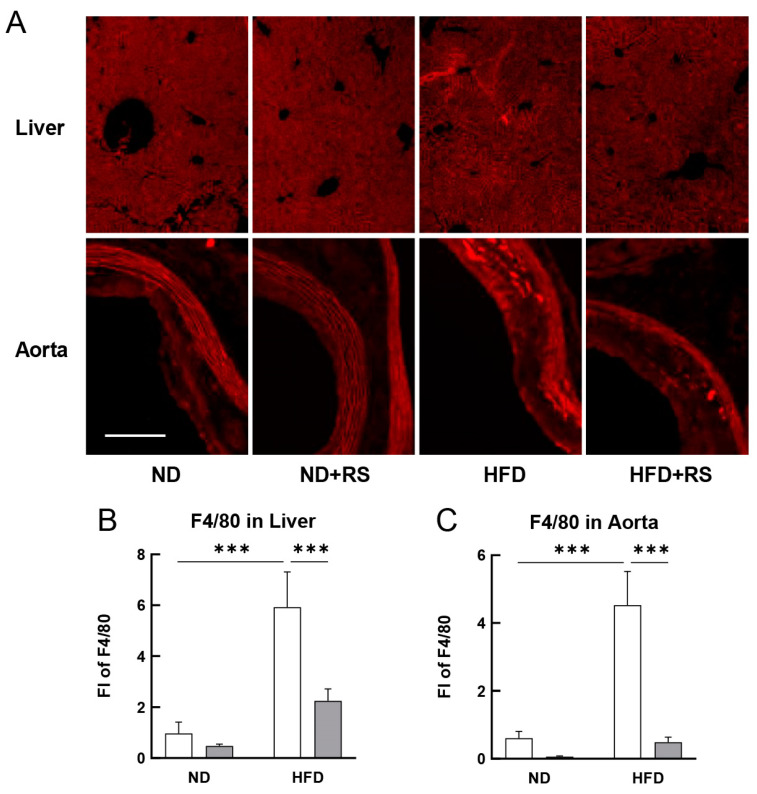
Effect of RS treatment on macrophage infiltration in the organ tissues of ApoE^−/−^ mice fed either an ND or HFD supplemented with or without RS. (**A**) Immunohistochemical analysis of F4/80 (marker of macrophage) in the liver and aorta of mice in the ND, ND + RS, HFD, and HFD + RS groups. Scale bar = 100 μm. Graphical representation of FI of F4/80 in the (**B**) liver and (**C**) aorta of mice in the ND and HFD groups. Open and gray columns indicate the RS-untreated and RS-treated groups, respectively. The data are shown as the mean ± SD. *** *p* < 0.001 indicates significant differences between each group using two-way ANOVA (n = 4).

**Figure 5 cells-12-02666-f005:**
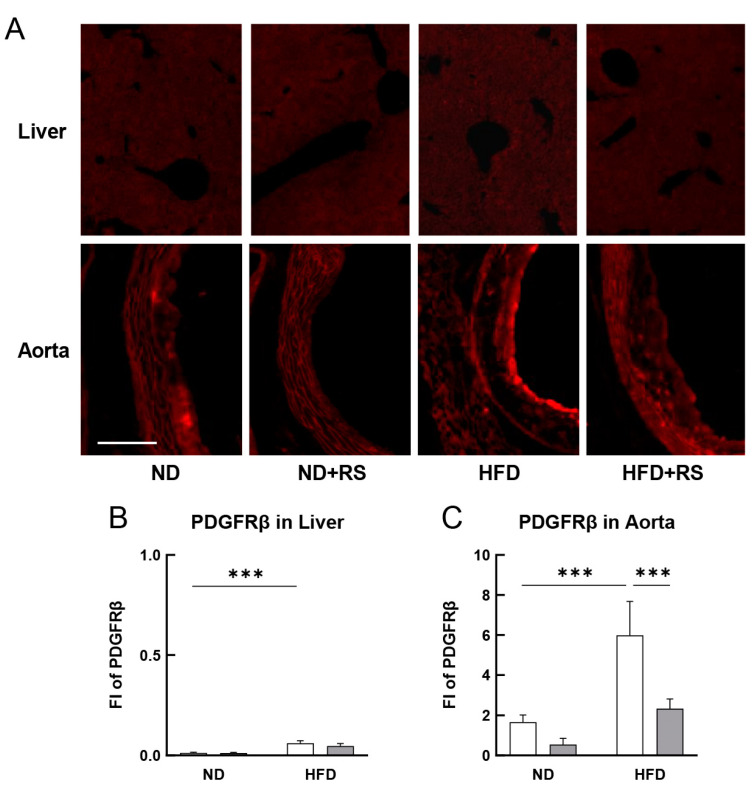
Effect of RS treatment on platelet-derived growth factor receptor (PDGFRβ) expression in the liver and aorta of ApoE^−/−^ mice fed an ND or HFD supplemented with or without RS. (**A**) Immunohistochemical analysis of PDGFRβ in the liver and aorta of mice in the ND, ND + RS, HFD, and HFD + RS groups. Scale bar = 100 μm. Graphical representation of FI of PDGFRβ in the (**B**) liver and (**C**) aorta of mice in the ND and HFD groups. Open and gray columns indicate the RS-untreated and RS-treated groups, respectively. The data are shown as mean ± SD. *** *p* < 0.001 indicates significant differences between each group using two-way ANOVA (n = 4).

**Figure 6 cells-12-02666-f006:**
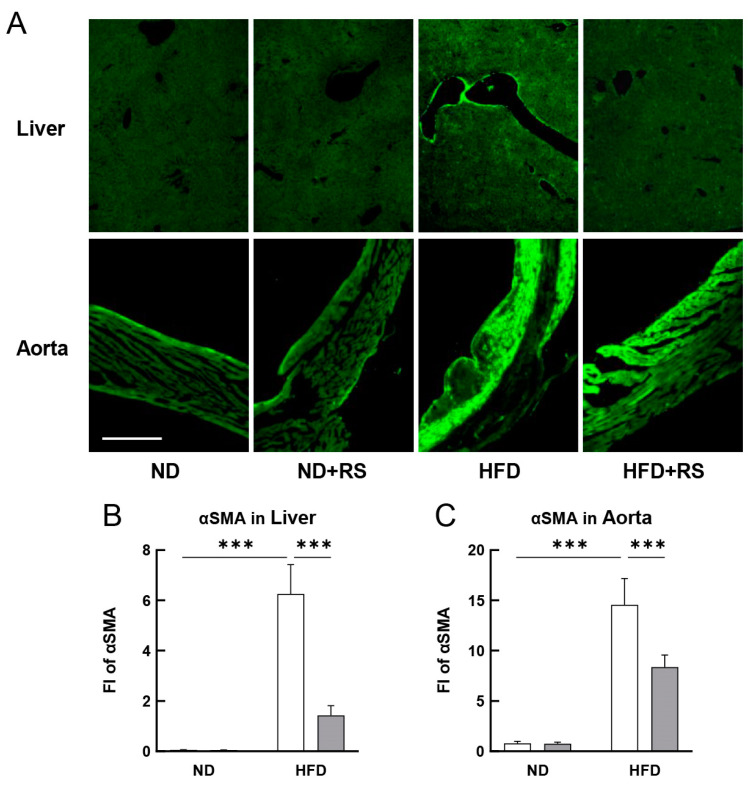
Effect of RS treatment on alpha smooth muscle actin (αSMA) expression in the liver and aorta of ApoE^−/−^ mice fed an ND or HFD supplemented with or without RS. (**A**) Immunohistochemical analysis of αSMA in the liver and aorta of mice in the ND, ND + RS, HFD, and HFD + RS groups. Scale bar = 100 μm. Graphical representation of FI of αSMA in the (**B**) liver and (**C**) aorta of mice in the ND and HFD groups. Open and gray columns indicate the RS-untreated and RS-treated groups, respectively. The data are shown as the mean ± SD. *** *p* < 0.001 indicates significant differences between the respective groups using two-way ANOVA (n = 4).

**Figure 7 cells-12-02666-f007:**
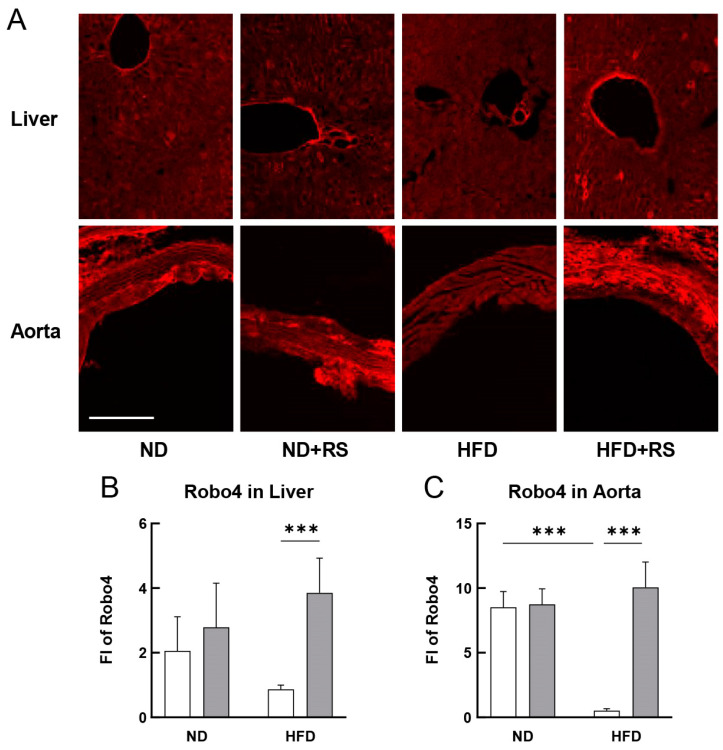
Effect of RS treatment on Roundabout 4 (Robo4) expression in the liver and aorta of ApoE^−/−^ mice fed an ND or HFD supplemented with or without RS. (**A**) Immunohistochemical analysis of Robo4 in the liver and aorta of mice in the ND, ND + RS, HFD, and HFD + RS groups. Scale bar = 100 μm. Graphical representation of FI of Robo4 in the (**B**) liver and (**C**) aorta of mice in the ND and HFD groups. Open and gray columns indicate the RS-untreated and RS-treated groups, respectively. Data are shown as the mean ± SD. *** *p* < 0.001 indicates significant differences between the respective groups using two-way ANOVA (n = 4).

**Figure 8 cells-12-02666-f008:**
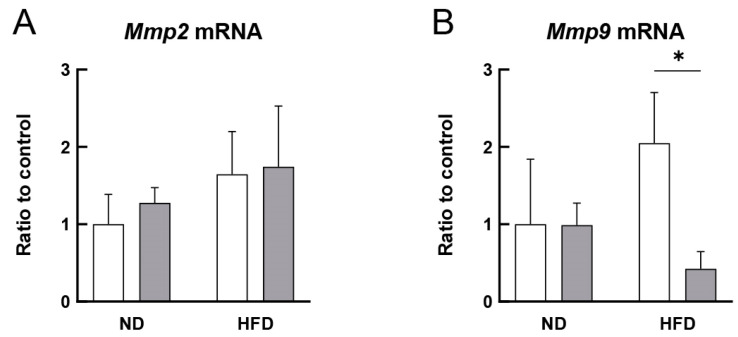
Effect of RS treatment on the mRNA expression of matrix metallopeptidase 2 (*Mmp2*) and *Mmp9* in the aorta of ApoE^−/−^ mice fed either an ND or HFD supplemented with or without RS. Graphical representation of (**A**) *Mmp2* and (**B**) *Mmp9* mRNA levels in the aorta of mice in ND, ND + RS, HFD, and HFD + RS groups, quantified with qPCR. Open and gray columns indicate the RS-untreated and RS-treated groups, respectively. Data are shown as the mean ± SD. * *p* < 0.05 indicates significant differences between the respective groups using two-way ANOVA (n = 3).

**Figure 9 cells-12-02666-f009:**
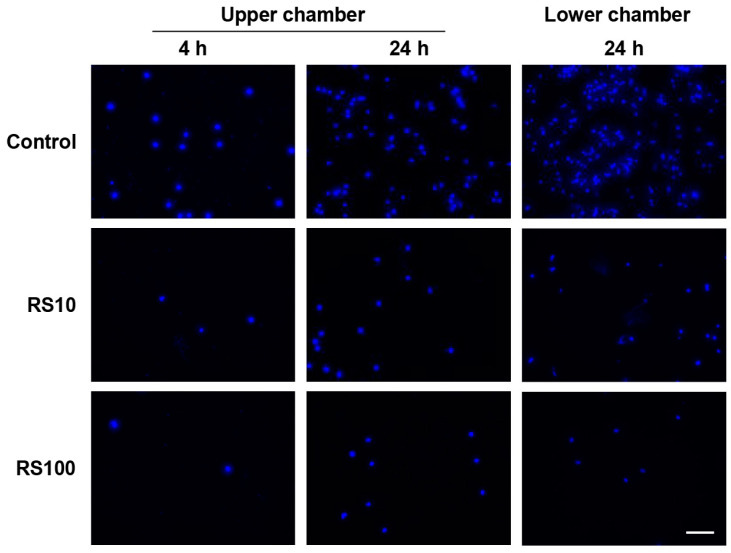
Effect of RS on the migration of RAW264.7 cells treated with or without RS. RS was added to the upper or lower chamber and incubated for 4 or 24 h. Cells that migrated to the underside of the membrane filter were fixed and stained with Hoechst33342 solution. Photographs (×200) of the representative image of cells stained with Hoechst are shown. Scale bar = 100 μm. The graphs on the left and in the center show the number of migrated cells at 4 and 24 h after placing RS in the upper chamber, respectively. The graph on the right shows the number of migrated cells 24 h after placing RS in the lower chamber. Data are shown as the mean ± SD. *** *p* < 0.001 indicates significant differences between the respective groups using one-way ANOVA (n = 5).

**Table 1 cells-12-02666-t001:** Pearson correlation coefficient (*r*) of each aorta-expressed molecule in relation to the plasma TCHO and plasma TG levels.

Aortic Molecules	Plasma TCHO		Plasma TG	
	*r*	*p*-Value	*r*	*p*-Value
VCAM-1	0.927	0.073	0.836	0.164
ICAM-1	0.932	0.068	0.562	0.438
F4/80	0.869	0.131	0.955	0.046 *
PDGFRβ	0.96	0.040 *	0.903	0.098
αSMA	0.958	0.042 *	0.692	0.308

* *p*-Value < 0.05 was statistically significant. Vascular cell adhesion molecule-1 (VCAM-1); intercellular adhesion molecule-1 (ICAM-1); platelet-derived growth factor receptor β (PDGFRβ); α smooth muscle actin (αSMA); total cholesterol (TCHO); triglyceride (TG).

## Data Availability

The data presented in this study are available upon request from the corresponding author.

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
