# Peer review of "Oral Administration of Rhamnan Sulfate from Monostroma nitidum Suppresses Atherosclerosis in ApoE-Deficient Mice Fed a High-Fat Diet"

_cells, 2023, doi:10.3390/cells12222666_

Round 1

Reviewer 1 Report

Comments and Suggestions for Authors

 Review Report 

Oral administration of rhamnan sulfate from Monostroma ni-tidum suppressed atherosclerosis in ApoE-deficient mice fed a high-fat diet.

This is a well-described study that shows the role of rhamnan sulfate (RS), as an effective functional food for the prevention of atherosclerosis.

1.       Can the authors please make a graphical representation of the study which will be helpful to the readers?

2.       Is it possible to show the study design in the format of a flow diagram which will be easy to understand?

3.       What other molecular-level techniques can be used to confirm the hypothesis mentioned in the manuscript Ex. Western blotting for specific proteins and RT-PCR for the specific genes.

4.         Please make all the figure legends in a uniform way. The Figure 1. heading is not bold and other figure headings are bold.

Author Response

For referee 1 comments

1)  Can the authors please make a graphical representation of the study which will be helpful to the readers?

Answer: Thank you for your suggestion. We will submit a graphical abstract based on our experimental results.

2)  Is it possible to show the study design in the format of a flow diagram which will be easy to understand?

Answer: Thank you for your suggestion. The experimental design is shown in Supplementary Figure S2.

3)  What other molecular-level techniques can be used to confirm the hypothesis mentioned in the manuscript Ex. Western blotting for specific proteins and RT-PCR for the specific genes.

Answer: Thank you for this suggestion. Our goal is to determine the biomolecules (proteins, etc.) that bind to RS or its metabolites and induce the expression of genes and proteins important for functional expression when RS is consumed. This analysis requires various techniques such as western blotting, RT-PCR, mass spectrometry, affinity chromatography, and so on. For example, we believe that it is necessary to predict molecular docking and conduct demonstration experiments using pull-down assays and affinity chromatography, as performed by Laeliocattleya et al. [50] in their research on fucoidan. Therefore, it is necessary to detect RS or its metabolites in mice. We are currently working on developing antibodies against RS, but that research could not completed on time.

[50] Laeliocattleya, R.A.; Yunianta, Y.; Risjani, Y.; Wulan, S.N. In silico molecular docking, molecular dynamics, ADMET analysis of fucoidan against receptor frizzled-8 and coreceptor LRP6 in Wnt/β-Catenin pathway and in vitro analysis of fucoidan extract from Sargassum echinocarpum as β-catenin inhibitor in breast cancer cell line (MCF-7). J. Biomol. Struct. Dyn. 2023, 1-16, doi: 10.1080/07391102.2023.2265488. 

4)  Please make all the figure legends in a uniform way. The Figure 1. heading is not bold and other figure headings are bold.

Answer: As per your suggestion, we have presented all figure headings with boldfacing for consistency.

Reviewer 2 Report

Comments and Suggestions for Authors

The authors investigated the anti-atherosclerotic effects of oral rhamnan sulfate administration in ApoE-deficient mice with high-fat diet. However, I have some comments.

1) The authors reported that oral rhamnan sulfate administration reduced vascular smooth muscle cell proliferation and macrophage accumulation in the aorta of ApoE-deficient mice. First of all, I recommend the authors to show the degree of the reduction in atherosclerotic plaque areas.

2) Oral rhamnan sulfate administration supressed the increase in blood total cholesterol and triglyceride levels. Was there any correlation between the reduction in vascular smooth muscle cell proliferation or macrophage accumulation and blood total cholesterol or triglyceride levels?

3) Please show the effect of oral rhamnan sulfate administration on inflammation of the aorta or blood inflammatory markers.

4) How many grams of monostroma nitidum for humans are equivant to oral rhamnan sulfate administration used in this experiment?

Author Response

For referee 2 comments

1)  The authors reported that oral rhamnan sulfate administration reduced vascular smooth muscle cell proliferation and macrophage accumulation in the aorta of ApoE-deficient mice. First of all, I recommend the authors to show the degree of the reduction in atherosclerotic plaque areas.

Answer: Thank you for your suggestion. Shortly after we started our research, a paper was published (Ref. 24 (Patil N.P. et al, Biomaterials 2022, 291, 121865.). In our preliminary study using HFD-fed ApoE-/- female mice, arterial histology showed that orally administered RS had almost the same anti-arteriosclerotic effect as described in the paper (Ref. 24). Unfortunately, due to the inexperience of the researchers, it was not possible to obtain clear histological pictures showing the extent of atherosclerotic plaque area reduction. Therefore, in this study, we decided to focus on the expression dynamics of atherosclerosis-related molecules in the liver and aorta. With this in mind, we have added the following text to the Introduction section (lines 90-99 pages 2-3):

Moreover, Patil et al. reported that RS reduces vascular inflammation and atherosclerotic plaque formation histologically in ApoE-/- female mice that are fed a high-fat diet (HFD) [24].

Therefore, in this study, we aimed to investigate the effects of oral administration of RS on blood lipids and vascular molecules involved in HFD-induced atherosclerosis in ApoE-/- female mice. The results showed that oral administration of RS ameliorated abnormal lipid metabolism and reduced atherosclerotic endothelial inflammation and hyperpermeability, macrophage infiltration and accumulation, and smooth muscle cell proliferation in the arteries of HFD-fed ApoE-/- mice. Furthermore, RS was found to directly inhibit the migration of macrophage-like RAW264.7 cells in vitro.

2)  Oral rhamnan sulfate administration suppressed the increase in blood total cholesterol and triglyceride levels. Was there any correlation between the reduction in vascular smooth muscle cell proliferation or macrophage accumulation and blood total cholesterol or triglyceride levels?

Answer: Thank you for your question. As per your comments, we determined the Pearson correlation coefficient (r) values between plasma total cholesterol (TCHO) and triglyceride (TG) levels and aorta atherosclerotic molecule levels in ApoE-/- mice fed ND or HFD with or without RS. The r values are shown in Table 1 under the newly added Results Section 3.7 (lines 410-430 page 12-13). Furthermore, the effect of RS administration on the correlation between changes in plasma TCHO and TG and changes in atherosclerotic aortic molecules is shown in Supplementary Figure S4.

Based on the Pearson Correlation coefficient values, the following text was added to the Discussion section (lines 550-559 page 17).

Pearson correlation coefficient data for each aorta-expressed molecule against changes in plasma TCHO and TG levels caused by oral administration of RS showed that changes in vascular cell molecules (PDGFRβ, αSMA, VCAM-1, and ICAM-1) correlated more with changes in plasma TCHO than those with plasma TG. However, changes in the macrophage-specific molecule F4/80 showed a higher correlation with changes in plasma TG than those with plasma TCHO. Saja et al. demonstrate that hypertriglyceridemia promotes monocytes extravasation and tissue macrophage accumulation [41]. Since high levels of TG-rich lipoproteins can promote macrophage formation into foam cells and plaque formation, lowering TG levels by RS administration may lead to inhibition of plaque formation.

[41] Saja, M.F.; Baudino, L.; Jackson, W.D.; Cook, H.T.; Malik, T.H.; Fossati-Jimack, L.; Ruseva, M.; Pickering, M.D.; Woollard, K.J.; Botto, M. Triglyceride-rich lipoproteins modulate the distribution and extravasation of Ly6C/Gr1(low) monocytes. Cell Rep. 2015, 12, 1802-1815, doi: 10.1016/j.celrep.2015.08.020.

3)  Please show the effect of oral rhamnan sulfate administration on inflammation of the aorta or blood inflammatory markers.

Answer: Thank you for your suggestion. In a previous paper (Ref. 19), we showed that the levels of inflammatory markers IL-6, tissue factor, von Willebrand factor, E-selectin, and VCAM-1 were significantly increased in the plasma of LPS-induced inflamed mice, and all marker levels were significantly decreased by oral administration of RS. Since RS administration decreased aortic VCAM-1 levels, which were increased in HFD-fed ApoE-/- mice, RS administration is presumed to reduce the levels of VCAM-1 and other inflammatory markers in the plasma of HFD-fed ApoE-/- mice.

4)  How many grams of Monostroma nitidum for humans are equivalent to oral rhamnan sulfate administration used in this experiment?

Answer: Thank you for your question. Calculating from the human equivalent dose (12.3 for mouse) based on body surface area [49], it is estimated that 0.73 g/day of rhamnan sulfate is required for a human weighing 60 kg, based on the experimental results using mice (body weight 30 g). Since dried Monostroma nitidum contains approximately 40% rhamnan sulfate [internal data, unreported], eating about 2 g/day of dried Monostroma nitidum may suppress atherosclerosis.

The following text was added to the Discussion section (lines 598-602 page 18):

Finally, based on current results, it can be estimated that consuming about 2 g of dried M. nitidum per day by a person weighing 60 kg may exert an anti-atherosclerotic effect, as calculated form the human equivalent dose based on body surface area (12.3 for mouse) [49] and the RS content of M. nitidum being approximately 40% [internal data, unreported].

[49] Nair, A.B.; Jacob, S. A simple practice guide for dose conversion between animals and human. J. Basic Clin. Pharma. 2016, 7, 27-31, doi:10.4103/0976-0105.177703

The complete text is as follows (lines 598-611 page 18):

Finally, based on current results, it can be estimated that consuming about 2 g of dried M. nitidum per day by a person weighing 60 kg may exert an anti-atherosclerotic effect, as calculated form the human equivalent dose based on body surface area (12.3 for mouse) [49] and the RS content of M. nitidum being approximately 40% [internal data, unreported]. From previous studies on fluorescein isothiocyanate-labeled RS, it is known that RS passes through M cells and enters the Peyer’s patches in the gastrointestinal tract after 30 min of oral administration [21]. However, it is unclear what substances RS or its metabolites bind to when consumed and whether they are involved in the expression of genes and proteins that are in turn involved in the expression of RS-dependent/RS-specific activities. This remains to be investigated in the future. Such analyses require various techniques to analyze RS and its effects of the body, for example, molecular docking prediction, analysis using RS-specific antibodies or radioisotope-labeled RS, and experiments such as pull-down assays and affinity chromatography, should be conducted, as performed by Laeliocattleya, et al [50] in their research on fucoidan.

[50] Laeliocattleya, R.A.; Yunianta, Y.; Risjani, Y.; Wulan, S.N. In silico molecular docking, molecular dynamics, ADMET analysis of fucoidan against receptor frizzled-8 and coreceptor LRP6 in Wnt/β-Catenin pathway and in vitro analysis of fucoidan extract from Sargassum echinocarpum as β-catenin inhibitor in breast cancer cell line (MCF-7). J. Biomol. Struct. Dyn. 2023, 1-16, doi: 10.1080/07391102.2023.2265488. 

Round 2

Reviewer 2 Report

Comments and Suggestions for Authors

I have no further comments.